# Adolescent–Parent Agreement on Callous–Unemotional Traits in Adolescents with Attention-Deficit/Hyperactivity Disorder

**DOI:** 10.3390/ijerph17113888

**Published:** 2020-05-30

**Authors:** Yi-Lung Chen, Ray C. Hsiao, Wen-Jiun Chou, Cheng-Fang Yen

**Affiliations:** 1Department of Healthcare Administration, Asia University, Taichung 41354, Taiwan; elong@asia.edu.tw; 2Department of Psychology, Asia University, Taichung 41354, Taiwan; 3Department of Psychiatry and Behavioral Sciences, University of Washington School of Medicine and Children’s Hospital, Seattle, WA 98105, USA; rhsiao@u.washington.edu; 4College of Medicine, Chang Gung University, Taoyuan 83332, Taiwan; 5Department of Child and Adolescent Psychiatry, Chang Gung Memorial Hospital, Kaohsiung Medical Center, 123, Dapi Road, Niaosong District, Kaohsiung 83325, Taiwan; 6Department of Psychiatry, Kaohsiung Medical University Hospital, No. 100, Tzyou 1st Road, Kaohsiung 80708, Taiwan; 7Department of Psychiatry, School of Medicine and Graduate Institute of Medicine, College of Medicine, Kaohsiung Medical University, Kaohsiung 80708, Taiwan

**Keywords:** attention-deficit/hyperactivity disorder, callous–unemotional traits, conduct disorder, cross-informant agreement, oppositional defiant disorder

## Abstract

This study examined the levels of agreement between the reports of 207 adolescents with attention-deficit/hyperactivity disorder (ADHD) and their parents regarding the adolescents’ callous–unemotional (CU) traits and investigated the factors influencing adolescent–parent agreement. Adolescent–parent agreement about CU traits in three dimensions according to the Chinese version of the Inventory of Callous and Unemotional Traits was examined. The influence of demographic characteristics, comorbid conduct disorder (CD), oppositional defiant disorder (ODD), and ADHD symptoms on adolescent–parent agreement was also examined. The results indicated that adolescent–parent agreement on the CU trait of uncaringness was moderate, whereas agreement on the CU traits of callousness and unemotionality was poor. Adolescent–parent agreement on the three dimensions of CU traits varied depending on the adolescents’ sex and comorbid CD and ODD symptoms as well as parental age. Therefore, multiple sources of information are required when assessing the severity of CU traits in adolescents with ADHD. The factors influencing the levels of the agreement should also be considered.

## 1. Introduction

### 1.1. Callous–Unemotional Traits and Mental Health

Callous–unemotional (CU) traits are a group of personality characteristics demonstrating a lack of care for values that others share, a lack of remorse, and a general poverty of affect [1,2]. Cross-sectional and prospective studies on children and adolescents in the community or clinical settings as well as forensic samples have reported that higher CU traits are positively associated with severe, stable, and aggressive patterns of antisocial behaviors [1,3,4,5,6], lower moral regulation, less guilt and empathy [7], more severe emotional problems [5], poorer peer functioning [8], and lower quality of life [9]. CU traits have also been used as a biomarker for research on brain and endocrine function, such as reward processing [10], executive function [11], and cortisol reactivity [12].

### 1.2. CU Traits in Children with Attention-Deficit/Hyperactivity Disorder

CU traits have unique roles in subgrouping, prognosis, and treatment responses in children and adolescents with attention-deficit/hyperactivity disorder (ADHD). Research on children with ADHD comorbid with oppositional defiant disorder (ODD) or conduct disorder (CD) determined that strong CU traits were associated with a lack of fearfulness, a reward-dominant response style, and a lack of regret for behavior problems [13]. CU traits were significantly associated with functional impairment in children with mild or moderate ADHD [14]. A review of 13 cross-sectional and retrospective studies indicated that CU traits in children with ADHD were a risk factor for later antisocial personality disorder [15]. Furthermore, a study determined that CU traits did not predict the remission of aggressive behaviors after stimulant treatment and family-focused behavioral intervention in children with ADHD and comorbid ODD or CD [16], whereas another study reported that CU predicted lower treatment responsiveness to behavioral therapy related to social skills, problem-solving, and negative behaviors in time-out among children with ADHD and comorbid CD [17]. The inconsistent findings indicated that further studies are needed to examine the roles of CU traits among children with ADHD from a clinical and research perspective.

### 1.3. Adolescent–Parent Disagreement on CU Traits

Studies have used reports of CU traits among adolescents from various informants, including adolescent self-reports [18,19], parent or legal guardian reports [4,5,9,20,21], and teacher or key-worker reports [22]. Few studies have examined adolescent–parent agreement on the severity of CU traits in adolescents. A study of 272 clinic-referred adolescents examined the level of adolescent–parent disagreement regarding CU trait levels using the Antisocial Process Screening Device (APSD) [23] and observed only a modest correlation between the self- and parent-reported CU traits of adolescents (*r* = 0.15, *p* < 0.05). They also revealed that self-reported anxiety levels were associated with behavioral and temperamental characteristics, whereas this relationship was not identified based on parent reports. A study on 91 nonreferred young adolescents determined that self-reports of CU traits on the APSD were moderately correlated with parent ratings [24]. Further studies examining adolescent–parent disagreement on CU traits may clarify the inconsistencies in findings from research regarding CU traits, considering that much of the related research has relied on self-report data [23]. Data from studies examining the level of adolescent–parent agreement on CU traits can also help clinicians and researchers assess whether they should collect information on CU traits from both adolescents and parents or rely on a sole informant.

### 1.4. Study Aims

The present study had two aims. First, CU traits contain various dimensions. The Inventory of Callous and Unemotional Traits (ICUT), one of the most used instruments, assesses three dimensions of CU traits in adolescents: callous, uncaring, and unemotional [25]. The dimensions of the ICUT capture various behavioral, affective, and cognitive CU traits [25]. For example, callousness and uncaringness among adolescents were more highly related to bullying than the trait of unemotionality [26]. Variances in adolescent–parent agreement across various dimensions of CU traits remain unclear. We hypothesized that adolescent–parent agreement levels correspond to various dimensions of CU traits in adolescents with ADHD.

Second, understanding the factors for predicting low levels of adolescent–parent agreement regarding CU traits is essential for the early detection of behavioral problems related to strong CU traits. Research has revealed that individuals with CD [27], ODD [28], or ADHD [29] had stronger CU traits compared with people without these psychiatric disorders. Men exhibited significantly higher CU levels compared with women [30]. Age is also critical in the development and stability of CU traits [24]. However, no study has examined adolescents’ and parental factors predicting the level of adolescent–parent agreement on CU traits in adolescents with ADHD. Therefore, we hypothesized that adolescents’ sex, age, and comorbid ODD, CD, and ADHD symptoms, and parents’ sex and education level, were associated with low adolescent–parent agreement regarding CU traits in adolescents with ADHD.

## 2. Methods

### 2.1. Participants

Adolescents with ADHD and their parents were enrolled from three child psychiatry outpatient clinics in Taiwan. Patients enrolled in Taiwan’s National Health Insurance program can visit outpatient clinics of teaching hospitals without referrals from general practitioners. Therefore, the adolescents enrolled from these clinics were representative of similar-age populations in Taiwan. The inclusion criteria for adolescents with ADHD were (1) an age range of 11–18 years and (2) a diagnosis of ADHD by a certified child psychiatrist according to the fifth edition of the *Diagnostic and Statistical Manual of Mental Disorders* [27]. The child psychiatrists reviewed patients’ medical records when they visited the outpatient clinics between October 2016 and July 2019. A total of 256 adolescents with ADHD who satisfied the inclusion criteria were consecutively approached in the outpatient clinics. The child psychiatrists interviewed adolescents and their parents to determine whether such adolescents had an intellectual disability, major psychiatric diseases (such as schizophrenia and bipolar disorder), or any other cognitive deficits that would cause difficulties in understanding the purpose of the study and completing research questionnaires. Based on the interview results, 22 adolescent–parent dyads were excluded. The purposes and procedures of the study were then explained to the 234 ADHD adolescent–parent dyads who satisfied the inclusion criteria; they were then invited to participate in the study. All possible participants were assured that their responses to the research questionnaire were confidential and that their participation or nonparticipation would not influence their right to receive medical services. A total of 207 (88.5%) ADHD adolescent–parent dyads agreed to participate in this study.

### 2.2. Measures

#### 2.2.1. The Chinese Version of the ICUT

The Chinese version of the ICUT consists of 24 items answered using a 4-point Likert scale to assess the level of how accurately items describe the participants, with 0, 1, 2, and 3 indicating “not at all”, “somewhat true”, “very true”, and “definitely true”, respectively [31]. This scale consists of three subscales: callousness (e.g., “I do not care whom I hurt to get what I want”), uncaring (e.g., “I seldom work hard on the things I do”), and unemotionality (e.g., “I do not show my emotions to others”). Higher total scores on the subscales indicate higher tendencies toward CU traits. A study examined the psychometrics of the C-ICUT in Chinese adolescents and reported that the C-ICUT has acceptable reliability and validity [32]. In the present study, both adolescent- and parent-reported CU traits on the C-ICUT of adolescents were obtained. The Cronbach’s α value ranges of the three subscales of the C-ICUT reported by adolescents and parents in the present study were 0.72–0.80 and 0.74–0.82, respectively.

#### 2.2.2. Short Form of the Swanson, Nolan, and Pelham Version IV Scale (SNAP-IV)-Chinese Version

The parent-reported Chinese-version short form of the Swanson, Nolan, and Pelham Version IV Scale (SNAP-IV-Chinese version) comprises 26 items rated on a 4-point Likert-like scale from 0 (not at all) to 3 (very much) for assessing the inattention, hyperactivity and impulsivity, and ODD symptoms of adolescents based on the criteria for ADHD and ODD specified in the fourth edition of the Diagnostic and Statistical Manual of Mental Disorders (DSM-IV) [33,34]. Higher total scores on the subscales indicate more severe ADHD and oppositional symptoms. Cronbach’s α values for inattention, hyperactivity and impulsivity, and ODD subscales were 0.91, 0.91, and 0.92, respectively.

#### 2.2.3. Psychiatric Comorbidity and Sociodemographic Characteristics

Adolescents were diagnosed as having CD and ODD based on clinical interviews and chart reviews. The sociodemographic characteristics of adolescents and their parents recorded in this study were sex (female or male), age, and parental duration of education (years).

### 2.3. Procedure

Adolescents with ADHD and their parents were invited to complete the research questionnaires in the interview rooms of outpatient clinics after providing written informed consent and being assured that their responses to the research questionnaire would be confidential. Two master’s-degree research assistants performed individual interviews with the adolescents to collect data on their self-reported CU traits. Before performing the research interviews, research assistants received comprehensive training on the application of research questionnaires. The principal investigator (C.-F.Y.) introduced the contents of the research questionnaires and discussed it with the research assistants. Each research assistant then performed a research interview with an adolescent with ADHD under the supervision of the principal investigator (C.-F.Y.) and received feedback for modification of the interview. Research assistants’ performance during the interviews was supervised regularly to assess fidelity. The parents completed the C-ICUT and short-form SNAP-IV. The parents could ask the research assistants for clarification if they had any questions about the questionnaires.

### 2.4. Ethics

All adolescents and their parents provided written informed consent. Parents also provided written informed consent regarding their children’s participation in this study. The study was conducted following the Declaration of Helsinki, and the protocol was approved by the Institutional Review Board of Kaohsiung Medical University (KMUHIRB-SV (I)-20150080).

### 2.5. Statistical Analysis

The descriptive results are presented as frequencies and percentages for categorical variables (i.e., adolescents’ and parents’ sex and psychiatric comorbidity) and as mean and standard deviation (SD) for continuous variables (i.e., adolescents’ and parents’ age; parental years of education; adolescents’ ADHD severity, oppositional symptoms, and CU traits).

We examined the level of agreement between adolescent- and parent-reported CU traits using a two-way random effect model, consistency, and intraclass correlation (ICC) for average measures. Negative ICC values (i.e., smaller than 0) were treated as 0 because negative values of ICC coefficients are not theoretically possible or meaningful [35]. According to Cicchetti [36], adolescent–parent agreement is poor, fair, good, and excellent if the ICC ranges are 0.00–0.39, 0.40–0.59, 0.60–0.74, and 0.75–1.00, respectively.

We categorized our participants based on adolescents’ and parents’ demographics as well as adolescents’ psychiatric comorbidity and ADHD symptoms to examine whether adolescent–parent agreement on CU traits differed according to adolescent and parent demographics and adolescents’ psychiatric comorbidities. For continuous variables (i.e., adolescent and parent ages, years of education for parents, adolescents’ ADHD and oppositional symptoms), the median score was used to dichotomize participants into low- and high-score groups.

To determine whether significant differences in ICC existed between the low- and high-score groups, Fisher’s transformation [37] was used, which converts the sampling distributions of the ICCs and their differences into a normal distribution. Fisher’s *z* test statistics and their corresponding *p* values were then computed.

## 3. Results

Table 1 presents the adolescents’ age, sex, CU traits, ADHD and ODD symptoms, and psychiatric comorbidities as well as parents’ age, sex, and years of education. Overall, 32 (15.5%) adolescents were girls and 175 (84.5%) were boys. The mean age of adolescents was 13.1 years (SD = 1.8 years). Regarding parents, 157 (75.8%) were women and 50 (24.2%) were men. The mean age of parents was 44.1 years (SD = 6.4 years). The mean year of education for parents was 13.5 years (SD = 2.7 years).

Table 2 summarizes the levels of agreement of adolescent–parent ratings regarding CU traits. The results indicated that adolescent–parent agreement on the traits of callousness and unemotionality was poor (ICC = 0.312 and 0.178, respectively), but agreement was moderate on the trait of uncaringness (ICC = 0.460).

The adolescent-reported and parent-reported scores of each item on the C-ICUT were further compared by paired *t* test with a *p* value < 0.002 (0.05/24) as statistically significant (Appendix A). The results demonstrated that adolescents reported a higher mean score on eight items than their parents, with the highest scores on the three items: “I always try my best”; “I work hard on everything I do”; and “I try not to hurt others’ feelings”. Parents reported a higher mean score on seven items than adolescents, with the highest scores on the three items: “The feelings of others are unimportant to me”; “I do not like to put the time into doing things well”; and “It is easy for others to tell how I am feeling”.

Table 3, Table 4 and Table 5 display the differences in levels of adolescent–parent agreement regarding adolescents’ three CU traits in various groups of adolescents with ADHD, categorized by adolescents’ and parents’ demographics and adolescents’ psychiatric comorbidities. For example, participants were divided into group 1 (G1, female) and group 2 (G2, male) according to adolescent’s sex. “G1 (*n*)” and “G2 (*n*)” indicated the number of participants in group 1 and group 2, respectively. “ICC in G1” and “ICC in G2” indicated the ICC of the CU traits in group 1 and group 2, respectively. For continuous variables (i.e., adolescent’s and parental ages, years of education for parents, adolescent’s ADHD and oppositional symptoms), the median score was used to dichotomize participants into low- and high-score groups.

The results indicated that adolescents without comorbid CD displayed lower adolescent–parent agreement on the trait of callousness (Table 3).

Adolescent–parent agreement regarding the trait of unemotionality was lower in adolescents with ADHD who had more ODD symptoms than in those who had less severe ODD symptoms (Table 4).

Girls, adolescents with comorbid CD, and adolescents with older parents displayed lower adolescent–parent agreement on the trait of uncaringness (Table 5).

## 4. Discussion

The present study on adolescents with ADHD determined that adolescent–parent agreement on the CU trait of uncaringness was moderate, whereas agreement on the CU traits of callousness and unemotionality was poor. Moreover, adolescent–parent agreement on the three dimensions of CU traits varied depending on the adolescents’ sex and comorbid CD and ODD symptoms and parental age.

### 4.1. Adolescent–Parent Agreement on Various Dimensions of CU Traits

The results indicated that adolescent–parent agreement on the trait of uncaringness was moderate, but the agreement on the traits of callousness and unemotionality was poor. The various characteristics of these three CU trait dimensions may partially account for the various adolescent–parent agreements regarding CU traits. The trait of uncaringness in the C-ICUT represents behaviors that reflect a lack of care regarding performance in tasks and the feelings of other people [25]. Adolescents with ADHD and a high level of uncaringness may view conforming to social expectations and pursuing school achievement as unnecessary in their growth process. These individuals may challenge authoritativeness, explore the values that convince them during adolescence, and form their emancipated self-identity [38]. Although challenging authoritativeness during adolescence is a normal developmental process and does not equate to the emergence of uncaringness, it provides parents with a basis for evaluating their level of uncaringness. Moreover, as a society that has been highly influenced by Confucianism, adolescents in Taiwan are encouraged to pursue social harmony and academic achievement [39]. Parents of adolescents may continuously perceive adolescents’ uncaringness with regard to social harmony and academic achievement from daily parenting and from the feedback of school staff, relatives, and neighbors.

The callousness trait captures an aspect of behavior that includes a lack of empathy, guilt, or remorse for misdeeds [25]. Unemotionality reflects an absence of emotional expression [25]. Although the traits of callousness and unemotionality may be disadvantageous to social harmony, they may be less frequently noticed than uncaringness in a system that focuses on achievement evaluation at school and in communities; therefore, parents may receive less negative feedback from others and are subsequently less aware of adolescents’ traits of callousness and unemotionality.

The present study did not collect information regarding adolescents’ CU traits from other sources and could not determine the accuracy of adolescent- and parent-reported CU traits. However, the low adolescent–parent agreement on CU traits indicates that multiple sources of information are required when clinicians and researchers assess CU traits in adolescents with ADHD.

### 4.2. Factors Related to Adolescent–Parent Disagreement on CU Traits

Previous studies mainly examined the CU traits and their predictive effects on prognoses in ADHD children and adolescents comorbid with CD [13,16,17]. The present study further examined the role of comorbid CD for the parent–adolescent agreement on the CU traits in adolescents with ADHD. The results of the present study demonstrated that adolescent–parent agreement on callousness was lower in ADHD adolescents without CD than in ADHD adolescents with CD whereas the level of adolescent–parent agreement regarding the trait of uncaringness was lower in ADHD adolescents with CD than that in those without CD. The DSM-5 lists “limited prosocial emotions” as one of the specifiers of CD [27]. Limited prosocial emotions and callousness are both characterized by a lack of remorse or guilt and a callous lack of empathy. Therefore, parents of adolescents may identify callousness more easily in adolescents with ADHD and CD [27]. Uncaringness is also a characteristic of “limited prosocial emotions” for the specifier of CD. The mechanism underlying these results warrants further study.

The present study revealed a lower level of adolescent–parent agreement regarding the unemotionality trait in ADHD adolescents with more severe ODD symptoms than in ADHD adolescents with less severe ODD symptoms. ODD refers to the persistent display of anger/irritability (e.g., losing temper, being touchy or easily annoyed, being angry and resentful), argumentation/defiant behaviors (e.g., arguing with authority figures, actively defying or refusing to comply with requests from authority figures or with rules, deliberately annoying others, and blaming others for his or her mistakes or misbehavior), and vindictiveness (e.g., being spiteful or vindictive) [27]. These emotional components of ODD are contrary to the presentation of the unemotional trait and may influence the judgment of parents on adolescents’ unemotionality.

The present study revealed that adolescent–parent agreement regarding the trait of uncaringness was lower among girls with ADHD and older parents compared with the agreement among boys with ADHD and younger parents, respectively. Studies have demonstrated that adolescent girls with ADHD have lower self-efficacy, poorer coping strategies, higher rates of depression and anxiety, and lower rates of physical aggression and other externalizing behaviors compared with adolescent boys with ADHD [40]. Furthermore, older parents may have more stringent concepts of behavioral norms compared with younger parents. The effects of sex difference in the presentation of ADHD and parental age differences in concepts of behavioral norms on adolescent–parent agreement regarding uncaringness warrant further study.

It is noteworthy that several studies have found poor parent–child agreement in the assessment of internalizing [41,42] and externalizing symptoms [43] in children. The present study relied on the information from the parents to determine children’s ADHD and ODD symptoms, which may influence the parent–children agreement on the CU traits.

### 4.3. Limitations

This study has limitations that should be addressed. First, no information on the CU traits of adolescents with ADHD was obtained from peers and teachers. Therefore, the possibility of comparing the level of agreement among various informants was limited. Second, the participants of this study were a clinical sample of adolescents with ADHD. Therefore, the results might not be generalizable to adolescents with ADHD in the community who have never received treatment. Moreover, adolescents and their parents who had an intellectual disability, major psychiatric diseases, or any other cognitive deficits with difficulties in understanding study purposes and completing research questionnaires were excluded. The results of the present study might not be generalizable to the groups of adolescents and parents who were excluded. Third, we did not survey parents’ CU traits and could not determine the influence of parental CU traits on adolescent–parent agreement.

## 5. Conclusions

The present study revealed that the level of adolescent–parent agreement depends on various dimensions of CU traits in adolescents with ADHD. Adolescent–parent agreement on the CU trait of uncaringness was moderate, whereas agreement on callousness and unemotionality were poor. The results indicate that clinicians and researchers should employ multiple sources of information when assessing CU traits in adolescents with ADHD, especially in groups of adolescents with ADHD who have characteristics that predict low adolescent–parent agreement, identified in this study.

## Figures and Tables

**Table 1 ijerph-17-03888-t001:** Levels of callous–unemotional traits, attention-deficit/hyperactivity disorder (ADHD) and oppositional defiant symptoms, and psychiatric comorbidities (*n* = 207).

Variables	*n* (%)/Mean ± SD
Callous–unemotional traits	
Callousness	
Adolescent-reported	8.9 ± 5.0
Parent-reported	12.1 ± 5.4
Unemotionality	
Adolescent-reported	7.5 ± 2.8
Parent-reported	5.9 ± 3.0
Uncaring	
Adolescent-reported	11.2 ± 4.5
Parent-reported	13.7 ± 4.3
SNAP-IV	
Inattention	13.8 ± 7.0
Hyperactivity/impulsivity	9.1 ± 6.5
Oppositional defiant	9.9 ± 6.4
Oppositional defiant disorder	115 (55.6%)
Conduct disorder	113 (54.6%)

SNAP-IV: the short version of the Swanson, Nolan, and Pelham Scale Version IV.

**Table 2 ijerph-17-03888-t002:** Adolescent–parent agreement ratings on callous–unemotional (CU) traits *n* = 207).

Variables	Intraclass Correlation
Callousness	0.312 **
Unemotionality	0.178
Uncaring	0.460 ***

** *p* < 0.01; *** *p* < 0.001.

**Table 3 ijerph-17-03888-t003:** Differences in the agreement of adolescent–parent ratings on the callousness trait in various groups of adolescents with ADHD.

	Groups				ICC of the Callousness Trait				
Variables	G1	G2	G1 (*n*)	G2 (*n*)	ICC in G1	ICC in G2	Difference in ICC between G1 and G2	*z*	*p*
Adolescent’s sex	Female	Male	32	175	0.088	0.356	−0.268	−1.415	0.157
Parental sex	Mother	Father	157	50	0.266	0.420	−0.154	−1.051	0.293
ODD	No	Yes	92	115	0.185	0.019	0.166	1.184	0.236
CD	No	Yes	94	113	0.084	0.350	−0.266	−1.985	0.047
Adolescent’s age ^a^	Low	High	136	71	0.287	0.361	0.074	0.555	0.579
Parental age ^a^	Low	High	105	102	0.372	0.243	0.129	1.016	0.312
Parental education ^a^	Low	High	145	61	0.348	0.263	0.085	0.604	0.547
Inattention ^a^	Low	High	105	102	0.200	0.177	−0.023	−0.169	0.866
Hyperactivity/impulsivity ^a^	Low	High	107	100	0.234	0.156	−0.078	−0.575	0.565
Oppositional defiant ^a^	Low	High	115	92	0.213	0.036	−0.177	−1.270	0.204

CD: conduct disorder; ICC: intraclass correlation; ODD: oppositional defiant disorder. ^a^ Participants were dichotomized into low- and high-score groups based on the median score. Median scores: adolescent’s age = 13; parental age = 44; parental education = 14; inattention = 13; hyperactivity–impulsivity = 8; oppositional defiance = 10.

**Table 4 ijerph-17-03888-t004:** Differences in the agreement of adolescent–parent ratings on the unemotionality trait in various groups of adolescents with ADHD.

	Groups				ICC of the Unemotionality Trait				
Variables	G1	G2	G1 (*n*)	G2 (*n*)	ICC in G1	ICC in G2	Difference in ICC between G1 and G2	*z*	*p*
Adolescent’s sex	Female	Male	32	175	0.445	0.111	0.334	1.828	0.068
Parental sex	Mother	Father	157	50	0.187	0.136	0.052	0.318	0.751
ODD	No	Yes	92	115	0.229	0.137	0.092	0.671	0.502
CD	No	Yes	94	113	0.221	0.113	0.108	0.785	0.433
Adolescent’s age ^a^	Low	High	136	71	0.109	0.268	0.159	1.109	0.268
Parental age ^a^	Low	High	105	102	0.154	0.215	−0.061	−0.449	0.654
Parental education ^a^	Low	High	145	61	0.225	0.077	0.148	0.974	0.330
Inattention ^a^	Low	High	105	102	0.177	0.183	0.006	0.044	0.965
Hyperactivity/impulsivity ^a^	Low	High	107	100	0.215	0.143	−0.072	−0.527	0.598
Oppositional defiant ^a^	Low	High	115	92	0.294	0.013	−0.281	−2.042	0.041

CD: conduct disorder; ICC: intraclass correlation; ODD: oppositional defiant disorder. ^a^ Participants were dichotomized into low- and high-score groups based on the median score. Median scores: adolescent’s age = 13; parental age = 44; parental education = 14; inattention = 13; hyperactivity–impulsivity = 8; oppositional defiance = 10.

**Table 5 ijerph-17-03888-t005:** Differences in the agreement of adolescent–parent ratings on the uncaring trait in various groups of adolescents with ADHD.

	Groups				ICC of the Uncaring Trait				
Variables	G1	G2	G1 (*n*)	G2 (*n*)	ICC in G1	ICC in G2	Difference in ICC between G1 and G2	*z*	*p*
Adolescent’s sex	Female	Male	32	175	0.000	0.514	−0.514	−2.830	0.005
Parental sex	Mother	Father	157	50	0.507	0.315	0.192	1.396	0.163
ODD	No	Yes	92	115	0.507	0.324	0.183	1.567	0.117
CD	No	Yes	94	113	0.536	0.304	0.232	2.008	0.045
Adolescent’s age ^a^	Low	High	136	71	0.387	0.577	0.190	1.675	0.094
Parental age ^a^	Low	High	105	102	0.575	0.337	0.238	2.161	0.031
Parental education ^a^	Low	High	145	61	0.498	0.386	0.111	0.890	0.371
Inattention ^a^	Low	High	105	102	0.495	0.272	−0.223	−1.869	0.062
Hyperactivity/impulsivity ^a^	Low	High	107	100	0.507	0.342	−0.165	−1.433	0.152
Oppositional defiant ^a^	Low	High	115	92	0.423	0.372	−0.051	−0.427	0.670

CD: conduct disorder; ICC: intraclass correlation; ODD: oppositional defiant disorder. ^a^ Participants were dichotomized into low- and high-score groups based on the median score. Median scores: adolescent’s age = 13; parental age = 44; parental education = 14; inattention = 13; hyperactivity–impulsivity = 8; oppositional defiance = 10.

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
