# Peer review of "Adolescent–Parent Agreement on Callous–Unemotional Traits in Adolescents with Attention-Deficit/Hyperactivity Disorder"

_ijerph, 2020, doi:10.3390/ijerph17113888_

Round 1

Reviewer 1 Report

The authors have done a study of callous unemotional (CU) traits to determine inter-rater agreement between self and parent report.   This is an original and clinically meaningful endeavor.  CU has become an important heuristic concept and a better understanding of the reliability of informants will contribute to better measurement as well as our conceptualization of the difficulty. 

Major Comments:

  1. The findings are interpreted mainly on the basis of the culture of Taiwan. It is also possible that the findings will hold up in cross cultural studies and reflect CU perceptions.   
  2. The authors should emphasize that the study further replicates other work demonstrating the overlap between the conceptual frameworks of CU and CD. 
  3. The discussion of ODD does not mention the symptom most pertinent to CU which is blaming others for your own mistakes. 
  4. The inter rater reliability is very interesting but more explanation is needed on how the authors interpret the degree of common perception on the basis of insight.  For this purpose the paper would be enriched by more information on who rated which symptoms as more severe. 

Minor comments:  

Page 2:  the review 1.2 notes findings that are not consistent with each other - this should be noted and some commentary provided. 

Author Response

Comment 1

The findings are interpreted mainly on the basis of the culture of Taiwan. It is also possible that the findings will hold up in cross cultural studies and reflect CU perceptions.

Response:

Thank you for your reminding. In the revised manuscript we rewrote the interpretations in Discussion section to avoid holding up the findings in cross cultural studies. Please refer to line 249-254.

“Adolescents with ADHD and a high level of uncaringness may view conforming to social expectations and pursuing school achievement as unnecessary in their growth process. These individuals may challenge authoritativeness, explore the values that convince them during adolescence, and form their emancipated self-identity [38]. Although challenging authoritativeness during adolescence is a normal developmental process and does not equate to the emergence of uncaringness, it provides parents with a basis for evaluating their level of uncaringness.”

Comment 2

The authors should emphasize that the study further replicates other work demonstrating the overlap between the conceptual frameworks of CU and CD.

Response

Thank you for your suggestion. We added the emphasis as below into Discussion section. Please refer to line 272-275.

Previous studies mainly examined the CU traits and their predictive effects on prognoses in ADHD children and adolescents comorbid with CD [13, 16, 17]. The present study further examined the role of comorbid CD for the parent-adolescent agreement on the CU traits in adolescents with ADHD.”

Comment 3

The discussion of ODD does not mention the symptom most pertinent to CU which is blaming others for your own mistakes.

Response

We added the symptom as below into the revised manuscript. Please refer to Discussion section, line 286-291.

“ODD refers to the persistent display of anger/irritability (e.g., losing temper, being touchy or easily annoyed, being angry and resentful), argumentation/defiant behaviors (e.g., arguing with authority figures, actively defying or refusing to comply with requests from authority figures or with rules, deliberately annoying others, and blaming others for his or her mistakes or misbehavior), and vindictiveness (e.g., being spiteful or vindictive) [27].”

Comment 4

The inter rater reliability is very interesting but more explanation is needed on how the authors interpret the degree of common perception on the basis of insight.  For this purpose the paper would be enriched by more information on who rated which symptoms as more severe. 

Response

Thank you for your suggestion. We added a supplementary table (Supplementary Table S1) to show the adolescent-reported and parent-reported mean scores on each item and the results of paired t test showing the information on who rated which symptoms as more severe. Please refer to Results section, line 204-211.

“The adolescent-reported and parent-reported scores of each item on the C-ICUT were further compared by paired t test with a p value <0.002 (0.05/24) as statistically significant (Supplementary Table S1). The results demonstrated that adolescents reported a higher mean score on 8 items than their parents, with the highest scores on the 3 items: “I always try my best.”; “I work hard on everything I do.”; and “I try not to hurt others' feelings.” Parents reported a higher mean score on 7 items than adolescents, with the highest scores on the 3 items: “The feelings of others are unimportant to me.”; “I do not like to put the time into doing things well.”; and “It is easy for others to tell how I am feeling.”

Comment 5

Page 2:  the review 1.2 notes findings that are not consistent with each other - this should be noted and some commentary provided.

Response

Thank you for your suggestion. We added a commentary as below for the results. Please refer to Results section, line 70-71.

“The inconsistent findings indicated that further studies are needed to examine the roles of CU traits among children with ADHD from a clinical and research perspective.”

Reviewer 2 Report

Dear authors, thank you very much for this interesting article that validates the Family Health Climate Scale for the Persian population.

Good article, you worked hard on this. It is relevant, as noted in the introduction, attention not only to children but also to families. The statement “a healthy society is made up of healthy families” is shared.

Below are some comments / questions and suggestions, personally.

  1. I would like understand the results but it was not possible because table 3 is unconfigured. I think this table provides a lot of information that I have not been able to assess, neither for the conclusions. However, this table is very extensive and I consider that it could be divided into three parts. I don't understand what you mean by n1 and n2, or the legend information.
  2. Regarding the results of Table 2, I consider the possible influence of the age of adolescents or if it is related to the information provided by the mother or father.
  3. Table 1 is also very broad and presents a lot of redundant information with the text.
  4. There are studies on the information provided by adolescents and their parents and the differences based on symptoms of internalization or externalization, I think it could be related to the results obtained.

Author Response

Comment 1

I would like understand the results but it was not possible because table 3 is unconfigured. I think this table provides a lot of information that I have not been able to assess, neither for the conclusions. However, this table is very extensive and I consider that it could be divided into three parts. I don't understand what you mean by n1 and n2, or the legend information.

Response:

Thank you for your comments. In the revised manuscript we divided the original Table 3 into three new tables (Table 3 for callousness, Table 4 for uncaring, and Table 5 for unemotionality5) to make the contents concise. We also revise the legend information. For example, “n1 and n2” was changed into “Group 1 and Group 2.” We added explanations as below for the legend information. Please refer to Results section, line 216-220.

For example, participants were divided into group 1 (G1, female) and group 2 (G2, male) according to adolescent’s sex. “G1 (n)” and “G2 (n)” indicated the number of participants in group 1 and group 2, respectively. “ICC in G1” and “ICC in G2” indicated the ICC of the CU traits in group 1 and group 2, respectively.”

Comment 2

Regarding the results of Table 2, I consider the possible influence of the age of adolescents or if it is related to the information provided by the mother or father.

Response:

We agree the reviewer’s opinion that the demographic characteristics of adolescents and parents may influence the adolescent–parent agreement on adolescents’ CU traits. We examined the influences of demographic characteristics on the three dimensions of CU traits in Table 3 to Table 5. Please refer to Results section, line 214-235.

Comment 3

Table 1 is also very broad and presents a lot of redundant information with the text.

Response:

Thank you for your comment. In the revised manuscript we deleted to information in the text if they have been presented in Table 1. Please refer to Results section, line 196.

Comment 4

There are studies on the information provided by adolescents and their parents and the differences based on symptoms of internalization or externalization. I think it could be related to the results obtained.

Response:

Thank you for your suggestion. In the revised manuscript we added the results of previous studies on low agreement of internalizing and externalizing behaviors in children between the reports of parents and children and the possible influence on the parent-children agreement of CU traits as below. Please refer to Discussion section, line 305-309.

“It is noteworthy that several studies have found poor parent-child agreement in the assessment of internalizing [41,42] and externalizing symptoms [43] in children. The present study relied on the information from the parents to determine children’s ADHD and ODD symptoms, which may influence the parent-children agreement on the CU traits.”

Round 2

Reviewer 1 Report

The authors have adequately addressed my concerns. 

Author Response

Thank you for your comments.

Reviewer 2 Report

The revision of the article shows important changes, explanatory comments and greater clarity in tables 3, 4 and 5, and some of the proposed questions have been answered. All the changes I think have improved the work, however there are unclear data and information not provided.

For example, in tables (3-5) they indicate that "participants were divided into group 1 (G1, female) and group 2 (G2, male) according to adolescent's sex", but this classification is not valid for other variables of the tables such as parents' age or socioeconomic level.
On the other hand, the criteria to indicate when variables such as adolescents' age, parents' age, parents' educational level ... have not been considered high or low.

Based on the results I don't understand the following comment "Adolescents with more ODD symptoms displayed lower adolescent–parent agreement on the 224 trait of unemotionality (Table 4)." (p224-225)

Regarding table 1, the redundant information has been removed, I would have removed information from such a long table and would have left it in the text, but this is a more personal criterion. Similarly, the information provided regarding the 4th comment is more appropriate in the discussion than in the limitations of the work presented.

Author Response

Comment

In tables (3-5) they indicate that "participants were divided into group 1 (G1, female) and group 2 (G2, male) according to adolescent's sex", but this classification is not valid for other variables of the tables such as parents' age or socioeconomic level. On the other hand, the criteria to indicate when variables such as adolescents' age, parents' age, parents' educational level ... have not been considered high or low.

Response

Thank you for your comment. For continuous variables (i.e., adolescent’s and parental ages, years of education for parents, adolescents’ ADHD and oppositional symptoms), the median score was used to dichotomize participants into low- and high-score groups. We added the explanation into Results section (line 221-224) and the footnotes of Tables 3 to 5 (line 230-232, 240-242, and 248-250).

a: Participants were dichotomized into low- and high-score groups based on the median score. Median scores: adolescent’s age = 13; parental age = 44; parental education = 14; inattention = 13; hyperactivity–impulsivity = 8; oppositional defiance = 10.”

Comment

Based on the results I don't understand the following comment "Adolescents with more ODD symptoms displayed lower adolescent–parent agreement on the 224 trait of unemotionality (Table 4)." (p224-225)

Response

We revised this sentence in to “Adolescent–parent agreement regarding the trait of unemotionality was lower in adolescents with ADHD who had more ODD symptoms than in those who had less severe ODD symptoms (Table 4).” Please refer to line 234-236.

Comment

Regarding table 1, the redundant information has been removed, I would have removed information from such a long table and would have left it in the text, but this is a more personal criterion.

Response

Thank you for your suggestion. We added back the participants information to the text and removed them from Table 1. Please refer to line 194-198 and Table 1.

Comment

Similarly, the information provided regarding the 4th comment is more appropriate in the discussion than in the limitations of the work presented.

Response

Thank you for your suggestion. In the revised manuscript we moved it from the Limitations section to Discussion section. Please refer to line 315-318.